# Rab44 Deficiency Induces Impaired Immune Responses to Nickel Allergy

**DOI:** 10.3390/ijms24020994

**Published:** 2023-01-04

**Authors:** Mayuko Noguromi, Yu Yamaguchi, Keiko Sato, Shun Oyakawa, Kuniaki Okamoto, Hiroshi Murata, Takayuki Tsukuba, Tomoko Kadowaki

**Affiliations:** 1Department of Frontier Oral Science, Graduate School of Biomedical Sciences, Nagasaki University, Sakamoto 1-7-1, Nagasaki 852-8588, Japan; 2Department of Dental Pharmacology, Graduate School of Biomedical Sciences, Nagasaki University, Sakamoto 1-7-1, Nagasaki 852-8588, Japan; 3Department of Prosthetic Dentistry, Graduate School of Biomedical Sciences, Nagasaki University, Sakamoto 1-7-1, Nagasaki 852-8588, Japan; 4Department of Dental Pharmacology, Graduate School of Medicine, Dentistry and Pharmaceutical Sciences, Okayama University, 2-5-1 Shikata-cho, Kita-ku, Okayama 700-8525, Japan

**Keywords:** Rab44, granulocytes, differentiation of blood cells, nickel allergy, immune responses

## Abstract

Rab44 was recently identified as an atypical Rab GTPase that possesses EF-hand and coiled-coil domains at the N-terminus, and a Rab-GTPase domain at the C-terminus. Rab44 is highly expressed in immune-related cells such as mast cells, macrophages, osteoclasts, and granulocyte-lineage cells in the bone marrow. Therefore, it is speculated that Rab44 is involved in the inflammation and differentiation of immune cells. However, little is known about the role of Rab44 in inflammation. In this study, we showed that Rab44 was upregulated during the early phase of differentiation of M1- and M2-type macrophages. Rab44-deficient mice exhibited impaired tumor necrosis factor alpha and interleukin-10 production after lipopolysaccharide (LPS) stimulation. The number of granulocytes in Rab44-deficient mice was lower, but the lymphocyte count in Rab44-deficient mice was significantly higher than that in wild-type mice after LPS stimulation. Moreover, Rab44-deficient macrophages showed impaired nickel-induced toxicity, and Rab44-deficient mice showed impaired nickel-induced hypersensitivity. Upon nickel hypersensitivity induction, Rab44-deficient mice showed different frequencies of immune cells in the blood and ears. Thus, it is likely that Rab44 is implicated in immune cell differentiation and inflammation, and Rab44 deficiency induces impaired immune responses to nickel allergies.

## 1. Introduction

Rab GTPases are critical regulators of intracellular membrane trafficking, which is essential for maintaining cellular functions [1,2,3]. Rab GTPases constitute the largest family of typical Ras-like proteins, including more than 70 members in humans, and regulate vesicular trafficking events by localizing to specific intracellular membranes and recruiting various types of Rab-effector proteins [4,5]. Rab GTPases are implicated in genetic disorders and diseases, including neurological diseases, cancer, immunological disorders, and infections [6,7,8]. Concerning immunological disorders, Rab27a gene mutations cause Griscelli syndrome, an autosomal disorder characterized by hypopigmentation of the skin and hair, reduced exocytosis of T-lymphocytes, and abnormal activation of macrophages [9,10]. The dislocation of basolateral proteins regulated by Rab13 results in Crohn’s disease, an inflammatory bowel disease characterized by chronic inflammation of the gastrointestinal tract [11]. Rab38 gene mutations lead to Hermansky–Pudlak syndrome, which is characterized by hypopigmentation and platelet storage deficiency associated with pulmonary fibrosis, granulomatous colitis, and immunodeficiency [12,13]. Distinct from these studies on “small” Rab GTPases, including Rab1-43, immune cell function of the “large” Rab GTPase Rab44 is poorly investigated.

Rab44 is a member of the large Rab GTPase family that encodes an N-terminal EF-hand domain, a coiled-coil domain, and a C-terminal Rab-GTPase domain with a molecular mass of approximately 110 kDa [14,15]. Previously, our research group had initially identified an upregulated Rab44 protein during osteoclast differentiation [16]. Rab44 knockdown enhances osteoclast differentiation, whereas Rab44 overexpression inhibits it [16]. In addition to osteoclasts, Rab44 is highly expressed in immune-related cells, such as mast cells and granulocyte-lineage cells, in the bone marrow [17]. Rab44 functions in mast cells’ degranulation. In fact, mast cells derived from Rab44-deficient mice show reduced histamine and lysosomal enzyme secretion compared with cells from wild-type mice in vitro [18]. Rab44-deficient mice also show reduced anaphylactic responses in vivo [19]. Given that Rab44 is highly expressed in immune-related cells, it is speculated that Rab44 is involved in the inflammation and differentiation of immune cells, in addition to mast cells. However, little information is available regarding these events.

Metals are widespread in our environment; however, they are often recognized as antigens inducing delayed-type hypersensitivity in vivo. Nickel (Ni) is the major cause of the allergic contact dermatitis as a symptom of delayed-type hypersensitivity. Although the detailed mechanism underlying metal allergies remains unexplored, histamine is considered to be involved in metal allergy. Histamine regulates immune responses associated with T cells, anaphylaxis, and physiological events such as wound healing and hematopoiesis [20,21,22].

In this study, we examined whether Rab44 expression changes during the differentiation of M1- and M2-type macrophages. Moreover, we found that Rab44-deficient mice exhibited impaired cytokine production and differential immune responses. We found that Rab44-deficient mice showed impaired contact hypersensitivity to nickel chloride.

## 2. Results

### 2.1. Rab44 Is Upregulated at an Early Phase during Differentiation of M1- and M2-Type Macrophages

We first examined the correlation between Rab44 expression and macrophage differentiation. In different microenvironments, macrophages are polarized into two subtypes: classically activated macrophages (M1) or alternatively activated macrophages (M2) [23,24]. We used the human monocytic cell line THP-1, which is useful for studying monocyte/macrophage differentiation [25,26], and examined the expression levels of Rab44 compared with M1 markers, including IL-6, CD80, and NOS2, and M2 markers, including interleukin-10 (IL-10), CD206, and arginase. A real-time PCR analysis of mRNA levels revealed that Rab44 was transiently upregulated at an early phase in both M1 (Figure 1a, left panel and b) and M2 differentiation (Figure 1a, right panel and b). Under M1 differentiation conditions, Rab44 sharply peaked prior to the expression of M1 markers and then gradually decreased (Figure 1b). During M2 differentiation, Rab44 quickly decreased after a sharp rise in the early phase (Figure 1b), and then the expression of M2 markers gradually increased. These results indicate that Rab44 is upregulated during the early phase of differentiation of M1- and M2-type macrophages.

### 2.2. Rab44-Deficient Mice Exhibit Impaired Cytokine Production and Differential Immune Responses

To examine the role of Rab44 in response to endotoxins, we analyzed cytokine production in mice in response to LPS. When wild-type and Rab44-knockout mice were injected intraperitoneally with LPS, the production levels of tumor necrosis factor alpha (TNF-α), a major inflammatory cytokine, and IL-10, a major anti-inflammatory cytokine, in the sera were measured (Figure 2). Rab44-knockout mice displayed lower but not significant cytokine levels of TNF-α and IL-10 compared to wild-type mice during 12–48 h after LPS stimulation (Figure 2a,b).

Next, we analyzed the profiles of white blood cell types of wild-type and Rab44-knockout mice at 48 h after LPS stimulation using a hematology counter (Figure 2c,d). The cell count and percentage of granulocytes in Rab44-knockout mice were significantly lower than those in wild-type mice (Figure 2c). Conversely, the percentage of lymphocytes in Rab44-knockout mice was significantly higher than that in wild-type mice (Figure 2d). There were no significant differences in monocyte numbers between the wild-type and Rab44-knockout mice (Appendix A). These results indicate that Rab44 deficiency affects white blood cell differentiation and proliferation in mice.

### 2.3. Rab44-Deficient Macrophages Show an Impaired Viability to Nickel Solution

Nickel is a major cause of allergic contact dermatitis by metals [27,28,29]. Therefore, we initially examined the effects of nickel on Rab44-knockout cells. Bone marrow-derived macrophages from wild-type and Rab44- knockout mice were exposed to various concentrations of nickel (Figure 3). The viabilities of Rab44-knockout macrophages were significantly higher than those of wild-type mice at low concentrations of less than 0.5 mg/mL NiSO_4_ (Figure 3). Thus, Rab44-knockout macrophages show an impaired sensitivity to nickel solutions.

### 2.4. Rab44-Deficient Mice Show Different Frequencies of Immune Cells upon Nickel-Hypersensitivity Induction

Next, we compared the effects of nickel on in vivo inflammatory responses in wild-type and Rab44-knockout mice. Several studies have shown that LPS promotes metal allergies in mice [28,29]. To sensitize mice to nickel, they were injected intraperitoneally with NiCl_2_ and a low concentration of LPS. Ten days after the injection, the mice were fed water supplemented with NiCl_2_ for 2 months to induce nickel allergy.

After 2 months feeding with nickel-water, wild-type mice showed hair loss on their backs, but Rab44-knockout mice hardly showed any change (Figure 4a). The IL-1β levels in the sera from nickel-water-fed wild-type mice were higher than those in Rab44-knockout mice, although the difference was not significant (Figure 4b). Previously it has been reported that IL-1β is required for the induction of nickel allergy through histamine production by histidine decarboxylase [29]. We further compared the white blood cell profiles between wild-type and Rab44-knockout mice using a hematology counter (Figure 4c,d). Similar to LPS stimulation, the number of granulocytes in Rab44-knockout mice was significantly lower than that in wild-type mice (Figure 4c). In wild-type mice, the percentage of granulocytes was slightly elevated by nickel-water feeding; however, it was decreased in Rab44-knockout mice. Moreover, the percentage of monocytes was significantly lower in Rab44-knockout mice than that in wild-type mice (Figure 4d).

### 2.5. Rab44-Deficient Mice Show Reduced Nickel-Induced Inflammation

After feeding nickel-water for 2 months, NiCl_2_ was injected into the root of each ear, and transient ear swelling was measured. The ear swelling during 0–24 h of the Rab44-knockout mice was significantly decreased compared to that of the wild-type mice (Figure 5).

To further investigate the distribution of immune cells, we performed fluorescent immunohistochemical staining (Figure 6). CD11b was used as a marker for monocytes, macrophages, and natural killer cells, whereas Gr-1 (Ly-6G) was used as a marker for granulocytes, such as neutrophils, eosinophils, basophils, and mast cells. We found that the number of CD11b-positive cells did not differ between Rab44-knockout and wild-type mice (Figure 6a,b). In contrast, Gr-1-positive cells were hardly observed in Rab44-knockout mice compared to those observed in wild-type mice (Figure 6b). However, there were no significant differences in the numbers of F4/80, a macrophage marker, and MHC-2, an antigen-presenting cell marker, between the wild-type and Rab44-knockout mice (Appendix A). These results indicate that Rab44-knockout mice show different frequencies of immune cells upon the induction of nickel hypersensitivity.

## 3. Discussion

In this study, we found that Rab44 was upregulated during the early phase of differentiation of M1- and M2-type macrophages. Rab44-deficient mice exhibited the impaired production of TNF-α and IL-10 after LPS stimulation. The granulocyte number in Rab44-deficient mice was lower, but the lymphocyte number in Rab44-deficient mice was significantly higher than that in wild-type mice after LPS stimulation. Rab44-deficient macrophages showed impaired nickel-induced toxicity, and Rab44-deficient mice showed impaired nickel-induced hypersensitivity. Thus, it is likely that Rab44 is implicated in immune cell differentiation and inflammation, and Rab44 deficiency induces impaired immune responses to nickel allergies.

Rab44 is likely to be involved in the differentiation of immune cells. In this study, Rab44 was found to be upregulated during the early phase of differentiation of M1- and M2-type macrophages. Similar results were observed in our previous study, which showed that Rab44 knockdown promotes the osteoclast differentiation of macrophages, whereas Rab44 overexpression prevents osteoclast differentiation [16]. Considering that Rab44 is highly expressed in myeloid progenitor cells in bone marrow cells, and the expression of Rab44 is downregulated in mature immune cells, except mast cells such as macrophages and neutrophils [17], it is speculated that Rab44 may be involved in autocrine secretion during the differentiation of various immune-related cells.

Decreased levels of TNF-α and IL-10 after LPS stimulation were observed in Rab44-knockout mice. Our previous study demonstrated that Rab44-knockout mice exhibited diminished anaphylaxis, and bone marrow mast cells derived from Rab44-knockout mice showed a decrease in FcεRI-mediated histamine and β-hexosaminidase secretion [19]. Therefore, Rab44 may regulate granule exocytosis in mast cells and IgE-mediated anaphylaxis in mice. Consistent with these results, the decreased cytokine production levels of TNF-α and IL-10 in Rab44-knockout mice after LPS stimulation could be due to the impaired section of various immune cells by Rab44 deficiency.

The decreased cytokine production in Rab44-knockout mice is probably associated with the phenotype that Rab44-knockout mice show impaired nickel hypersensitivity. In Rab44-kockout mice, nickel-induced stimulation strongly affected Gr1-positive cells but did not greatly affect CD11-positive cells. These results suggest that Gr1-positive cells often coincide with Rab44-expressing cells, whereas CD11b-positive cells cover approximately 44% of myeloid cells, overlapping only a small fraction of the Rab44-expressing cell population. To date, the detailed molecular mechanisms remain unknown. However, a recent study has also reported that Rab44 affects immune cell subpopulations. Krayem et al. [30] identified a unique mouse strain, B10.O20, in the spleen, which showed a significantly higher frequency of myeloid-derived cells, such as CD11b^+^, CD11b^+^Gr1^+^, CD14^+^, and F4/80^+^ cells than other strains of mice. In that study, a bioinformatics analysis of genomic sequences indicated that Rab44 was a candidate gene for influencing the frequencies of immune cell subpopulations, since the mouse strain displayed different expression levels of Rab44 mRNA and a Rab44 gene mutation G275R, in which glycine at the position of the Rab44 gene was changed to arginine [30]. Therefore, it is likely that the Rab44 expression levels are involved in frequencies of immune-cell subpopulations in mice. Furthermore, nickel-water-fed mice showed elevated serum levels of IL-1β; however, the upregulation of IL-1β was attenuated in Rab44-knockout mice compared to that in wild-type mice. It has been reported that nickel allergy is attenuated in IL-1β-deficient mice [29]. Impaired IL-1β induction in Rab44-knockout mice may be due to defects in the differentiation of immune cells.

## 4. Materials and Methods

### 4.1. Antibodies and Reagents

Phorbol 12-myristate 13-acetate (PMA), IFN-γ, First-grade NiCl_2_, NiSO_4_, and mouse M-CSF were purchased from FUJIFILM Wako Pure Chemicals (Osaka, Japan), and interleukin (IL)−4, IL-13, and IFN-γ were purchased from PeproTech. The concentrations of the cytokines TNF-α or IL-10 in the mouse sera were determined using enzyme-linked immunosorbent assays (Proteintech). An anti-CD11b antibody and anti-Gr-1 (Ly-6G) antibodies were purchased from Proteintech. Alexa Fluor 488-conjugated goat anti-rabbit IgG and Alexa Fluor 555-conjugated goat anti-mouse, anti-rat, and anti-rabbit IgG were obtained from Thermo Fisher Scientific. Other reagents, including the protease inhibitor cocktail and LPS were purchased from Sigma Aldrich. THP-1 cells were purchased from the American Type Culture Collection.

### 4.2. Animals

Wild-type and Rab44-knockout mice with a C57BL/6 genetic background were used, as described previously [19]. All experiments were performed with age-matched female wild-type and Rab44-knockout mice littermates. All animal experimental protocols were approved by the Animal Care and Use Committee of the Nagasaki University Graduate School of Biomedical Sciences (Permit 1703071365 and 2107211733).

### 4.3. Cell Culture

THP-1 cells were cultured in a complete RPMI 1640 medium containing 10% FBS, penicillin (50 U/mL), and streptomycin (50 µg/mL). To initially induce M0 macrophage differentiation, THP-1 cells were incubated with 0.1 μM PMA in a complete RPMI 1640 medium for 24 h. They were further cultured for 24 h with 0.1 μg/mL LPS and 20 ng/mL IFN-γ in a complete RPMI 1640 medium to induce M1 macrophage differentiation, or further cultured for 24 h with 20 ng/mL IL-4 and 20 ng/mL IL-13 in a complete RPMI 1640 medium to induce M2 macrophage differentiation. Bone marrow-derived macrophages were isolated according to a previously described method [31]. Briefly, marrow cells from mice femurs and tibias were cultured overnight in an α-modified eagle minimum essential medium (MEM) containing 10% fetal bovine serum (FBS), penicillin (100 U/mL), and streptomycin (100 μg/mL) at 37 °C in 5% CO_2_. Non-adherent cells were harvested in a stroma-free bone marrow cell culture system containing 50 ng/mL M-CSF. After 3 days, the adherent cells were harvested as bone marrow-derived macrophages.

### 4.4. Reverse Transcription-Polymerase Chain Reaction (RT-PCR) Analysis

An RT-PCR analysis was performed as described previously [32]. Briefly, total RNA was extracted from the tissues or cells using TRIzol reagent (Invitrogen, Carlsbad, CA, USA). cDNA was obtained by reverse transcription using Revertra Ace (Toyobo, Osaka, Japan). A quantitative RT-PCR was performed using the TB Green Premix Ex Taq II (Takara Bio) and Quant Studio 3 (Thermo Fisher Scientific, Waltham, MA, USA). The following primer sets were used.

Mouse GAPDH, forward: AACGACCCCTTCATTGACCTC and reverse:ACTGTGCCGTTGAATTTGCC; mouse Rab44, forward: AGAGACCACACACACTCTC and reverse: CTCCTGTAAGTCTGTTCTTG; IL-6, forward: ACTCACCTCTTCAGAACGAATTG reverse: CCATCTTTGGAAGGTTCAGGTTG; CD80, forward: CTCTTGGTGCTGGCTGGTCTT reverse: GCCAGTAGATGCGAGTTTTGTGC;

NOS2, forward: ACAAGCCTACCCCTCCAGAT reverse: TCCCGTCAGTTGGTAGGTTC; IL-10, forward: GGCTACGGCGCTGTCATCGATT reverse: GCATTCTTCACCTGCTCCACGG; CD206, forward: AGCCAACACCAGCTCCTCAAGA reverse: CAAAACGCTCGCGCATTGTCCA; Arginase, forward: GGCTGGTCTGCTTGAGAAAC reverse: CTTTTCCCACAGACCTTGGA.

### 4.5. Cell Viability to Nickel Solution

Macrophages seeded in 96-well culture plates were rinsed with PBS, incubated with different concentrations of nickel solution for 24 h at 37 °C, and measured with the Cell Counting Kit-8 solution (Dojindo, Kumamoto, Japan) for 1 h. Absorbance at 450 nm was measured with a microplate reader (Multiskan FC, Thermo Fisher Scientific, Waltham, MA, USA).

### 4.6. Nickel-Induced Allergic Contact Dermatitis

Experimental nickel-induced allergic contact dermatitis was performed essentially according to the method described by Kinbara et al. [20], with some modifications. Briefly, female mice were sensitized by intraperitoneal injection with 250 μL of 1μg/mL LPS and 1 mM NiCl_2_ solution. Ten days after sensitization, the mice were fed with the 0.03 mg/mL NiCl_2_ solution for 2 months. After induction, the mice were injected with 20 μL of 1 mM NiCl_2_ solution into both auricles of the mice, and ear-swelling was measured using a Peacock dial thickness gauge (Ozaki MFG Co., Ltd., Tokyo, Japan).

### 4.7. Immunofluorescence Microscopy

Immunohistochemistry was performed as previously described [17]. Briefly, the tissues were fixed with 4.0% paraformaldehyde in PBS for 20 min at 25 °C, and then washed with distilled water. After quenching with 50 mM NH_4_Cl for 10 min, the samples were washed with 0.1% Tween 20 in PBS for 10 min, and then with PBS three times. The samples were incubated with primary antibodies at 4 °C overnight after blocking with 1.0% bovine serum albumin in PBS. The samples were washed four times with PBS and then incubated with Alexa Fluor-conjugated secondary antibodies, followed by nuclear staining with DAPI. The samples were observed using an all-in-one microscopic imaging system (BZ-X800, KEYENCE, Osaka, Japan).

### 4.8. Statistical Analysis

Quantitative data are presented as mean ± standard error (S. E.). Statistically analyses were performed using JMP Pro 15 (SAS Institute, Tokyo, Japan). The unpaired *t*-test was used to identify differences between concentrations when a significant difference (* *p* < 0.05 or ** *p* < 0.01) was determined by the analysis of variance.

## 5. Conclusions

In conclusion, Rab44 deficiency induces impaired immune responses to nickel allergy.

## Figures and Tables

**Figure 1 ijms-24-00994-f001:**
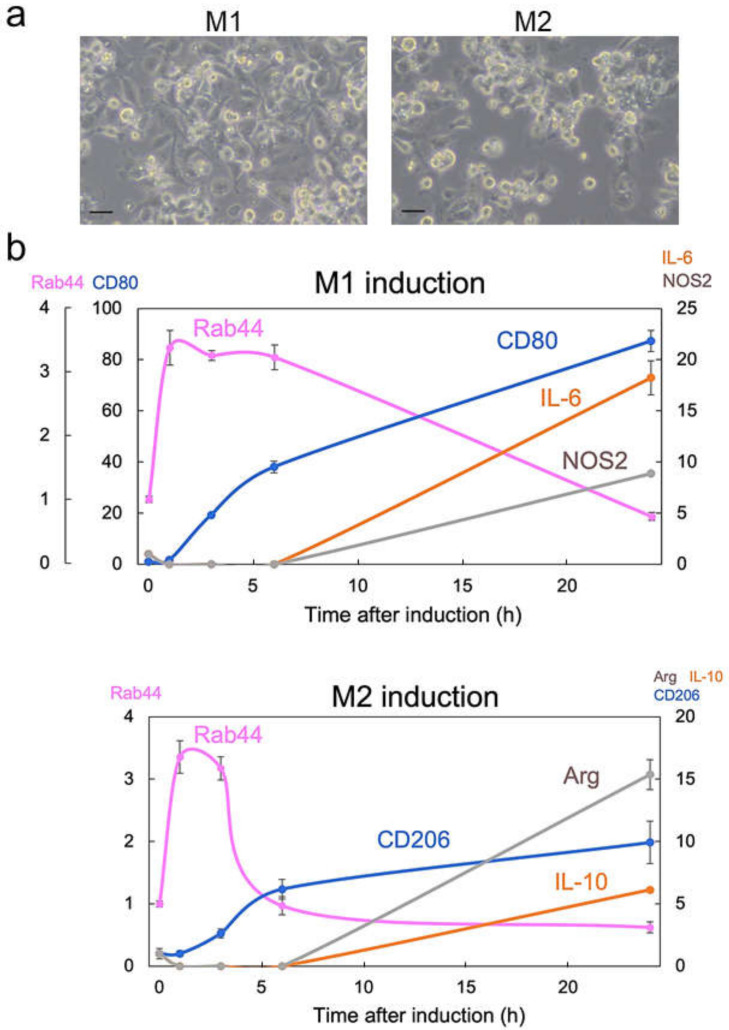
Upregulation of Rab44 in macrophage differentiation. THP-1 cells were induced to M0 macrophages by treatment with phorbol 12-myristate 13-acetate (PMA) (100 nM) at 37 °C for 24 h. M0 macrophages were differentiated into M1 macrophages by incubation with lipopolysaccharide (LPS) (0.1 mg/mL) and interferon (IFN)-γ (20 ng/mL), or M2 macrophages by incubation with IL-4 (20 ng/mL) and IL-13 (20 ng/mL) at 37 °C for 0–24 h. (**a**) Representative images of M1- and M2-type macrophages are shown. Bars: 20 μm. (**b**) Quantitative RT-PCR determination of Rab44 mRNA expression in M1- or M2-type macrophages for the indicated times. Data are represented as the mean ± S.E. of values from three independent experiments.

**Figure 2 ijms-24-00994-f002:**
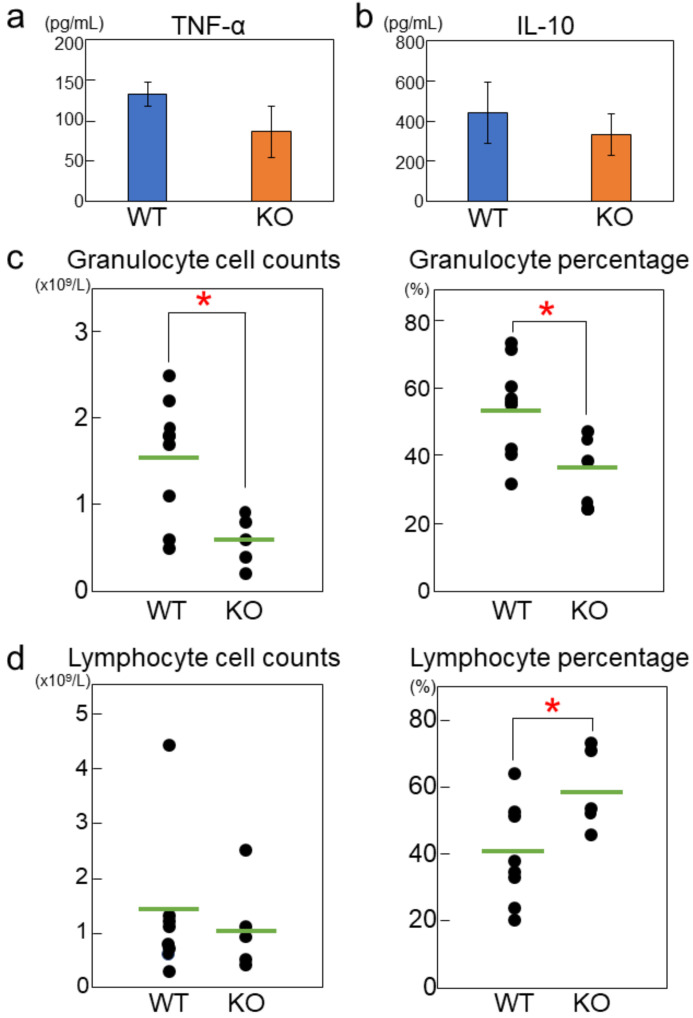
Different immune responses between wild-type (WT) and Rab44-knockout mice (KO) to LPS administration. (**a**,**b**) WT and Rab44-KO female mice were intraperitoneally injected with 0.2 mg of LPS (O55:B5). After the injection, blood was collected from the mandibular or tail vein at 12 h. The cytokine levels of TNF-α (**a**) and IL-10 (**b**) in the sera were measured using ELISA. (**c**,**d**) After 48 h of injection, the blood was analyzed with a hematology counter, Thika (ARKRAY) (WT, n = 8; KO, n = 5). * *p* < 0.05 compared between WT and KO mice. Granulocytes (**c**) and lymphocytes (**d**) counts (left panels), and percentages in white blood cells (right panels).

**Figure 3 ijms-24-00994-f003:**
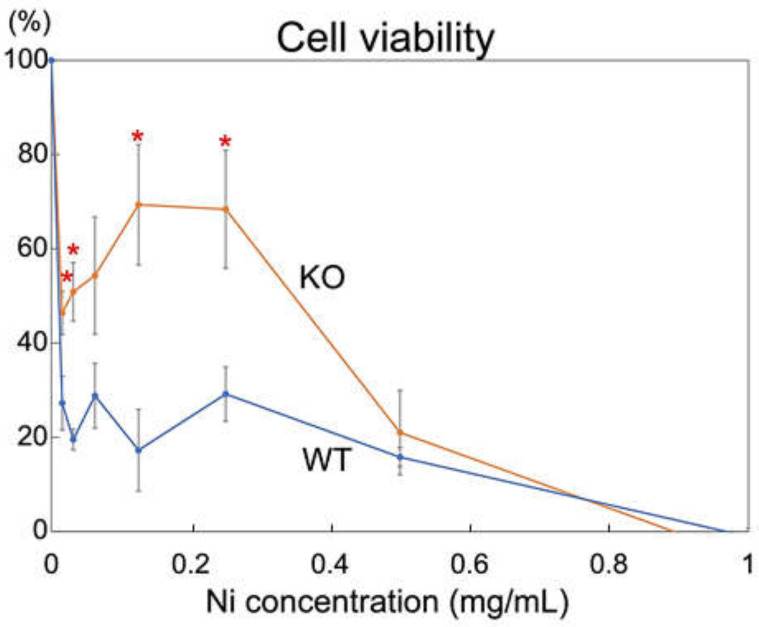
Sensitivity of macrophages to nickel solution. Bone marrow-derived macrophages derived from wild-type (WT) and Rab44-knockout (KO) mice were incubated with the indicated concentrations of Ni_2_SO_4_ solution for 24 h. Viability was determined using a Cell Counting Kit. * *p* < 0.05 compared to WT mice.

**Figure 4 ijms-24-00994-f004:**
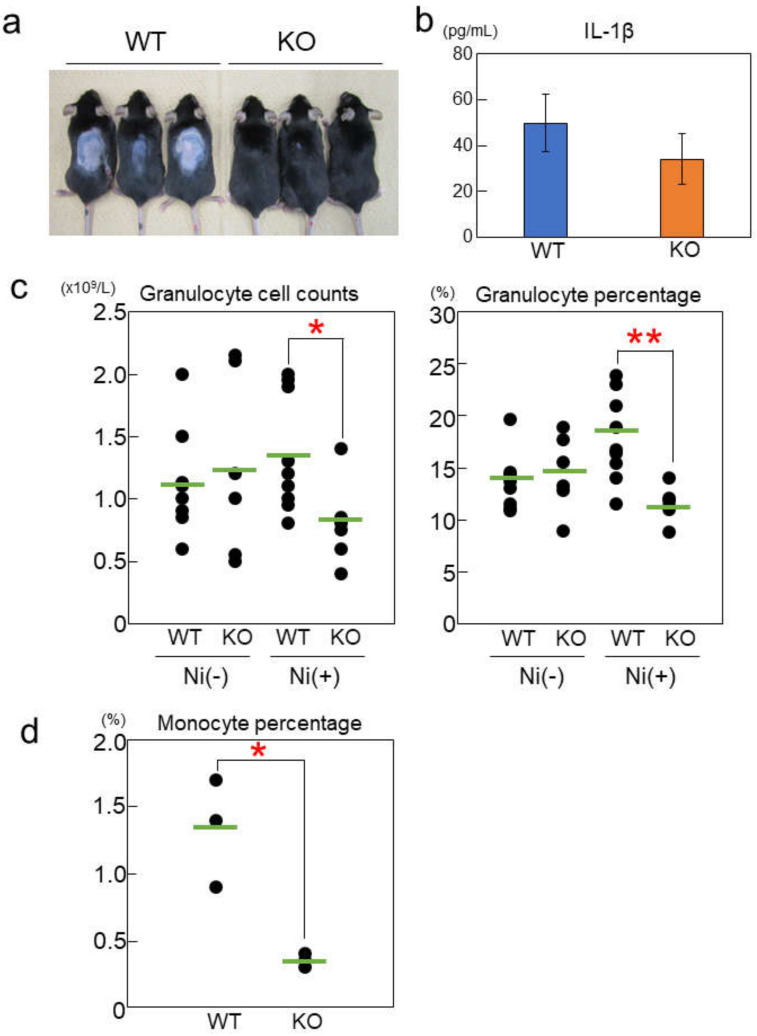
Responses of mice fed with nickel-water. (**a**) Gross appearance of wild-type (WT) and Rab44-knockout (KO) mice fed with nickel-water for 2 months. (**b**) Serum IL-1β level of WT and Rab44-KO mice fed with nickel-water for 2 months. (**c**) Granulocytes counts and percentage in white blood cells of WT and Rab44-KO mice before or after administration of nickel-water. (**d**) Monocyte percentage in white blood cells of WT and Rab44-KO mice fed with nickel-water for 2 months. * *p* < 0.05 and ** *p* < 0.01 compared between WT and KO mice.

**Figure 5 ijms-24-00994-f005:**
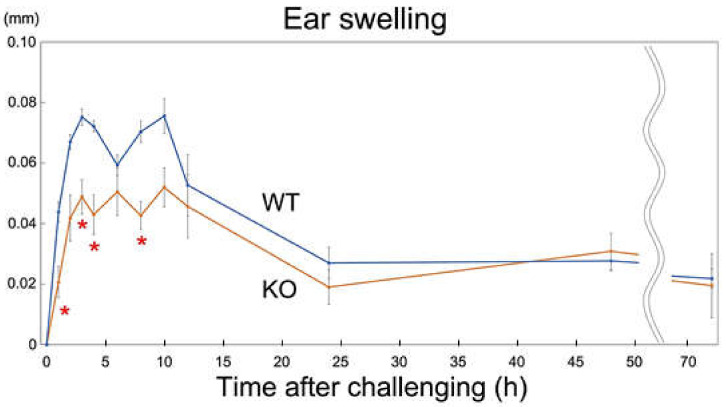
Ear swelling of mice in response to nickel solution. Wild-type (WT) and Rab44-knockout (KO) mice were sensitized with LPS and a nickel solution. Ten days after sensitization, the mice were fed with nickel-water for 2 months. Two months later, the mice were injected with the nickel solution. Ear thickness was measured before the stimulation and for 72 h after the stimulation. * *p* < 0.05 compared to WT.

**Figure 6 ijms-24-00994-f006:**
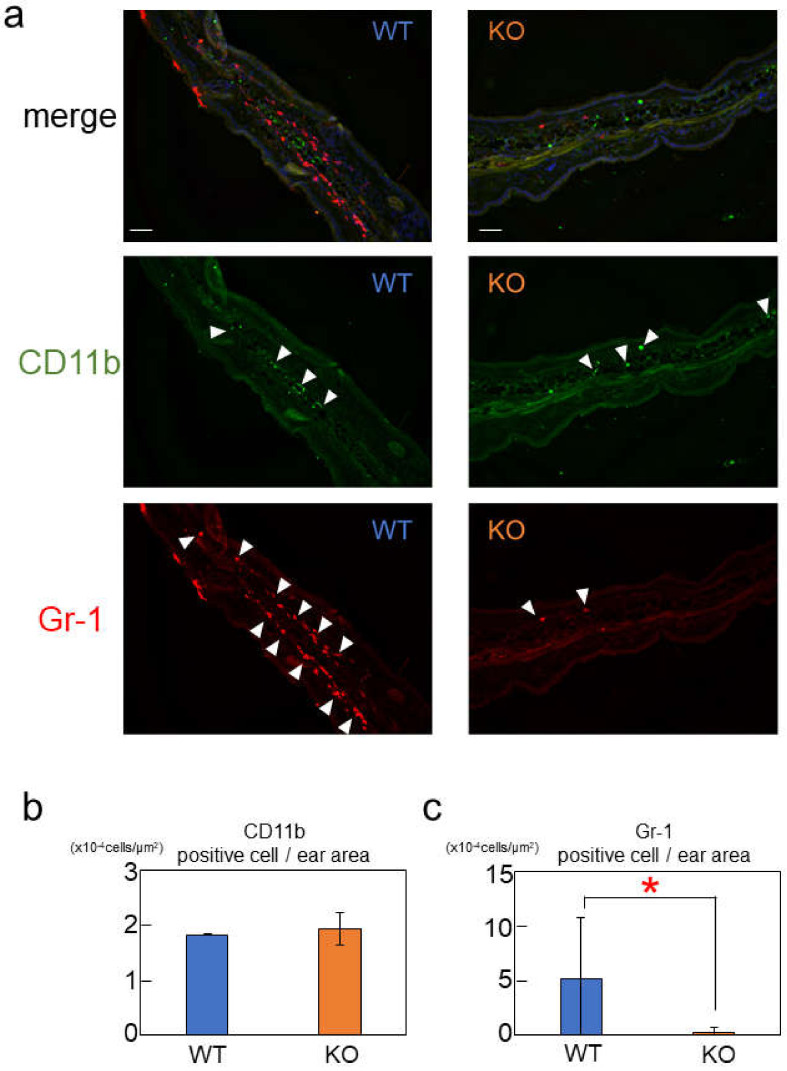
Immunofluorescence analysis of CD11b and Gr-1-positive cells in nickel-induced ear in wild-type (WT) and Rab44-knockout (KO) mice. (**a**) The fixed sections of the nickel-treated ear were blocked with 1.0% bovine serum albumin in PBS. The samples were incubated with rabbit anti-CD11b IgG (1:400) and rat anti-Gr-1 IgG (1:200), as the primary antibody, followed by fluorescent labelling with Alexa Fluor 488-conjugated anti-rabbit IgG and Alexa Fluor 555-conjugated anti-rat IgG, and then observed by microscopy. Bars: 50 μm. White arrowheads indicate the antibody reactive cells. (**b**,**c**) Quantitative analysis of CD11b and Gr-1-positive cells per μm^2^ after visualization by microscopy. Data are represented as the mean ± S.E. of values from three independent experiments. Asterisks indicate statistical significance compared to the control; * *p* < 0.05.

## Data Availability

The data presented in this study are available upon request from the corresponding author.

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
