# Peer review of "Rab44 Deficiency Induces Impaired Immune Responses to Nickel Allergy"

_ijms, 2023, doi:10.3390/ijms24020994_

Round 1

Reviewer 1 Report

Overall, the manuscript makes a good impression and is interesting. There are several inaccuracies in it. The part describing the methodology describes the performance of more types of research than it results from the description of the results obtained. I mean that there is a description of the method and there are no results of the described test. Thus, the methodological part requires a thorough review and editing, so that it is consistent with the described results. Other comments are minor typographical errors (e.g. line 160 says "wide-type" when it should be "wild-type").

Author Response

Reviewer 1

Overall, the manuscript makes a good impression and is interesting. There are several inaccuracies in it. The part describing the methodology describes the performance of more types of research than it results from the description of the results obtained. I mean that there is a description of the method and there are no results of the described test. Thus, the methodological part requires a thorough review and editing, so that it is consistent with the described results. Other comments are minor typographical errors (e.g. line 160 says "wide-type" when it should be "wild-type").

Answer: First of all, we appreciate for positive comments for our paper. The comment is quite right. According to the reviewer’s suggestion, we have deleted some words of “Materials and Methods”. Meanwhile, according to the other reviewer’s suggestion, we added the Animal Welfare Committee approval number for animal experiments in 4.2 Animals of “Materials and Methods”.

Moreover, thank you for suggesting the typographical error (e.g. line 160 says "wide-type" when it should be "wild-type"). We corrected it.

Reviewer 2 Report

This study explored the effect of Rab44 deficiency on the immune response to nickel allergy, which is a comprehensive and rigorous study and worthy of publication. However, this manuscript remains a handful of mistakes. I suggest that there should be a minor revision before it is accepted for publication.

Comment 1: Please upload the results not shown in result 2.2 as supplementary data.

Comment 2: Please confirm whether there is no significant difference in the number of CD11b wild-type and Rab44-knockout positive cells in Figure 6.

Comment 3: Please discuss why CD11b is not significantly different.

Comment 4: Please upload MHC-2 fluorescent immunohistochemical staining results as: Other data.

Comment 5: The Animal Welfare Committee approval number for animal experiments does not appear in the manuscript, please add it now.

Author Response

Reviewer 2.

This study explored the effect of Rab44 deficiency on the immune response to nickel allergy, which is a comprehensive and rigorous study and worthy of publication. However, this manuscript remains a handful of mistakes. I suggest that there should be a minor revision before it is accepted for publication.

Answer: We first appreciate for your positive comments. Thank you for some important comments.

Comment 1: Please upload the results not shown in result 2.2 as supplementary data. Answer: According to the reviewer’s suggestion, we added the data of monocyte numbers between the wild-type and Rab44-knockout mice as Supplementary Figure S1.

Comment 2: Please confirm whether there is no significant difference in the number of CD11b wild-type and Rab44-knockout positive cells in Figure 6.

Answer: Thank you for your important comments. We confirmed that there is no significant difference in the number of CD11b wild-type and Rab44-knockout positive cells.

Comment 3: Please discuss why CD11b is not significantly different.

Answer: According to the reviewer’s suggestion, we added the description why the number of CD11b-positive cells did not differ between Rab44-knockout and wild-type mice, on lanes 239-243, such as “In Rab44-kockout mice, nickel-induced stimulation strongly affected Gr1-positive cells but did not greatly affect CD11-positive cells. These results suggest that Gr1 positive cells often coincide with Rab44-expressing cells, whereas CD11b-positive cells cover approximately 44% of myeloid cells, overlapping only a small fraction of the Rab44-expresing cell population.

Comment 4: Please upload MHC-2 fluorescent immunohistochemical staining results as: Other data.

Answer: Thank you for the useful comment. According to the reviewer’s suggestion, we added the data of MHC-2 fluorescent immunohistochemical staining in Supplementary Figure S2.

Comment 5: The Animal Welfare Committee approval number for animal experiments does not appear in the manuscript, please add it now.

Answer: Thank you for the important comment. We added the Animal Welfare Committee approval number for animal experiments on lanes 271-276 as follows. “4.2. Animals Wild-type and Rab44-knockout mice with a C57BL/6 genetic background were used as described previously [19]. All experiments were performed with age-matched female wild-type and Rab44-knockout mice littermates. All animal experimental protocols were approved by the Animal Care and Use Committee of Nagasaki University Graduate School of Biomedical Sciences (Permit 2107211733).”